# Demonstration of a fast and easy sample-to-answer protocol for tuberculosis screening in point-of-care settings: A proof of concept study

Nasir Ali[1,2], Graziele Lima Bello[3,4], Maria Lúcia Rosa Rossetti[3,4], Marco Aurelio Krieger[1,2,5], Alexandre Dias Tavares Costa[1,5] *

1 Instituto de Biologia Molecular do Paraná (IBMP), Curitiba, Paraná, Brazil, 2 Departamento de Engenharia de Bioprocessos e Biotecnologia, Universidade Federal do Paraná (UFPR), Curitiba, Paraná, Brazil, 3 Centro de Desenvolvimento Científico e Tecnológico, Secretaria Estadual de Saúde, Porto Alegre, Rio Grande do Sul (RS), Brazil, 4 Programa de Pós-Graduação em Biologia Celular e Molecular Aplicado à Saúde (PPGBiosaúde), Universidade Luterana do Brasil, Canoas, Rio Grande do Sul, Brazil, 5 Instituto Carlos Chagas (ICC), Fundação Oswaldo Cruz (FIOCRUZ), Curitiba, Paraná, Brazil

* alexandre.costa@fiocruz.br

**Data Availability Statement:** All relevant data are within the paper and its Supporting Information files. Some data cannot be shared publicly because they might contain patient information. However,

## Abstract

We sought to develop a smooth and low cost sample preparation and DNA extraction protocol, streamlined with a ready-to-use qPCR in a portable instrument to overcome some of the existing hurdles. Several solutions were evaluated as to their ability to liquefy a mucin-based matrix. Each liquefied matrix, supplemented with either *Mycobacterium tuberculosis* (MTB) H37Rv strain DNA or intact cells, was aliquoted onto a filter paper embedded with solubilizing agents, and was subsequently dried up. Most of the nucleic acids, including genomic DNA from the bacilli and the host, binds to the filter paper. Next, several protocols were evaluated to elute the DNA from the paper, using qPCR to detect the insertion sequence IS6110, a *M. tuberculosis* complex genomic marker. The limit of detection (LOD) of the best protocol was then evaluated using parallel seeding and colony counting. The protocol was also evaluated using seventeen sputum samples, previously characterized by the GeneXpert or culture. Two instruments (the ABI7500 Standard and the Q3-Plus system) and two reagents storage formats (frozen or ready-to-use) were evaluated. Solutions containing guanidine isothiocyanate exerted the best liquefying effect on the mucin-based matrix extracted from one 6-mm punches, followed by a brief incubation at 95°C. The resulting DNA contained impurities, but a simple 1:10 dilution elicited the detection of MTB and human genomic targets. The described protocol presented an apparent LOD of 02 CFU/mL of MTB. Challenging the protocol with previously characterized samples showed substantial agreement with GeneXpert MTB/RIF results (sensitivity of 90%, agreement of 88.9%, kappa coefficient of 0.77), and moderate agreement with culture results (sensitivity of 100%, agreement of 78.9%, kappa coefficient of 0.58). This work presents a sensitive proof–of–concept protocol for sputum liquefaction and decontamination followed by a simple DNA extraction procedure, in which the extraction steps are streamlined with a ready-to-use qPCR in a portable instrument that can be employed in low infrastructure settings.

these data can be obtained upon request by contacting the Ethics Committee of the Universidade Luterana do Brasil at the email comitedeetica@ulbra.br.

**Funding:** Several authors received grants or fellowships from the Brazilian funding agencies CNPq and CAPES. This work was funded by a grant from the Banco Nacional de Desenvolvimento Econômico e Social (BNDES), contract no. 15.2.0473.1 (Operation #4.816.864), and by a grant from the Fundação de Amparo à Pesquisa do Rio Grande do Sul (FAPERGS grant #17/2551-0001542-8). There was no additional external funding received for this study. IBMP, BNDES, FAPERGS, CAPES or CNPq had no participation in the present study's design, data collection, analysis or interpretation, or writing of the report and decision to submit for publication.

**Competing interests:** Instituto de Biologia Molecular do Paraná (IBMP) produces the PCR mastermix used in the qPCR experiments, and NA, MAK, and ADTC are associated with IBMP. This does not alter our adherence to PLOS ONE policies on sharing data and materials.

## Introduction

Tuberculosis (TB), caused by the bacillus *Mycobacterium tuberculosis* (MTB), is one of the leading causes of mortality from a single infectious microorganism, which disproportionately affects low and middle-income countries. In 2016, 1.7 million people have died and 10.4 million been fell ill due to TB, while 56% of the cases were from only five countries: India, Indonesia, China, Philippines and Pakistan [1]. Many diagnostic tests already exist, but a large gap of 4.1 million people without proper TB diagnosis still poses a serious global health challenge [2]. TB diagnosis is routinely performed by sputum culturing, smear microscopy, chest x-rays, the tuberculin skin test, or nucleic acid-based molecular tests [3]. Although nucleic acid amplification and detection techniques such as PCR and LAMP have open a new era of specific and sensitive diagnostic testing, these tests are mostly available in developed countries and resource rich areas. Reasons for the lack of widespread use are many, but costs, cumbersome sample preparation and nucleic acid extraction protocols as well as fragile instruments are the most prominent ones [3, 4].

Sample preparation has always been a challenge for molecular assays as well as a bottleneck in translating complex molecular based tests to easy-to-use point of care products [4]. Moreover, the sputum samples used in TB diagnosis pose additional challenges, such as the probability of being highly contagious as well as its high viscosity. Sputum culture is an important diagnostic tool in TB control programs, because it is more sensitive than smear microscopy and facilitates drug susceptibility testing, even though its results take too long to obtain. However, because sputum samples pass through the oropharynx during collection, culture contamination limits the diagnostic yield of sputum culture for TB. To overcome this problem, the World Health Organization (WHO) standard procedure for sputum decontamination and liquefaction recommends mixing the sample with a solution containing N-acetyl-L-cysteine and sodium hydroxide (NALC/NaOH solution) for no more than 15–20 minutes [5, 6]. WHO sample treatment principles are two: (1) the short exposure to NaOH kills all bacteria present in the sputum sample with the exception of *M. tuberculosis*, and (2) the mucolytic agent NALC reduces the disulfide bonds of the extracellular matrix components of the sputum, liquefying the high protein sample, thus allowing easier manipulation and the use of lower concentrations of NaOH.

Chaotropic agents such as urea or guanidine (hydrochloride or isothiocyanate) are known to disrupt non-covalent bonds within biological molecules, such as those that hold together proteins tertiary structure or lipid bilayers. Indeed, chaotropic agents have long been used to disrupt cellular structures for nucleic acid extraction in molecular biology [4] or to increase protein extraction yield in biochemical protocols [7]. Interestingly, high concentrations of guanidine hydrochloride have been shown to inactivate *M. tuberculosis* bacilli in sputum samples following exposure to common lysis buffers contained in nucleic acid extraction kits [8].

Most of the molecular assays developed to date depend on dedicated laboratory space, complex protocols, as well as highly sophisticated and delicate instruments. Although some tests have been developed for use in low infrastructure settings, their costs are still prohibitive for mass screening in countries where tuberculosis is endemic. A rapid, simple, efficient, less expensive and ideally equipment-free sample preparation would not only bring down the cost of molecular assays but also mitigate the challenge of access to low-resource settings [9, 10]. Furthermore, the reagents used in molecular assays require special storage and transportation, which further increase their costs [11, 12]. Largely, the successful integration of sample preparation with ready-to-use reagents and portable qPCR systems would solve the main hurdles in point of care testing [9, 10, 13].

We sought to develop a procedure that uses the whole sputum sample (avoiding sample heterogeneity) to screen for the *IS*6110 insertion element sequence of *Mycobacterium* complex

[14]. We evaluated several chaotropic agents for their ability to liquefy sputum-like porcine mucin-based samples, which are common substitutes for sputum in research settings [15]. The liquefied mixture was stored in a detergent-containing commercial filter paper, and several subsequent steps for DNA extraction were evaluated. Extracted DNA was assayed for the presence of *M. tuberculosis* DNA using ready-to-use qPCR reagents preloaded onto a portable thermocycler. Lastly, the full protocol, from sample liquefaction and filter paper DNA extraction to detection by qPCR, was evaluated using a small number of patient samples, and results were then compared to those obtained by a different molecular method (GeneXpert MTB/RIF) as well as culture.

The proof of concept protocol presented here is a simple and straightforward diagnostic solution that could be employed to screen MTB-suspected patients in low-resource areas, and the positive results warrant future larger studies.

## Materials and methods

### Clinical specimens

Sputum samples were collected from new patients attending the Department of Thisiology and Leprosy of the city of Canoas (RS, Brazil) in two consecutive days according to WHO and the Brazilian Ministry of Health recommendations [16]. Inclusion criteria were no previous treatment and ability to produce a minimal sample volume per day (500 μL). Exclusion criteria were previous treatment and inability to produce enough sample volume. Day 01 samples were analyzed by culture and the GeneXpert MTB/Rif assay. The remaining volume of day 01 sample was pooled with day 2 sample and 500 μL were used for the evaluation of the new paper-based DNA extraction protocol. Non-TB samples were defined by negative results on the GeneXpert MTB/RIF assay and culture. The Ethics Committee for Research in Humans of the Universidade Luterana do Brasil (Canoas, Brazil) approved the present study. Samples were part of the diagnostic routine of the Department of Thisiology and Leprosy and would be discarded. Therefore, the ethics committee did not require informed consent from the patients (CAAE: 70697116.7.0000.5349, #1.786.666). A general workflow of the whole procedure is presented in Fig 1.

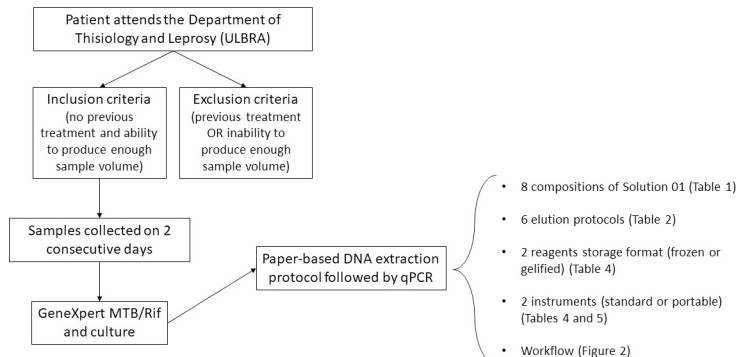

**Fig 1. Schematic representation of the experimental workflow.** Patients attending the Department of Thisiology and Leprosy of the Universidade Luterana do Brasil (ULBRA) were included in the study if had not had previous TB treatment and were able to produce a minimum of 500 μL of sputum per collection day. Accordingly, patients were excluded if had previous TB treatment or were unable to produce the minimal sample volume. Samples were collected on two consecutive days and were subjected to the GeneXpert MTB/Rif assay and culture. Next, eight liquefying solutions and six elution protocols were evaluated using the same samples. Eluted DNA were evaluated by two qPCR instruments and two reagents' storage formats.

## Solutions used in optimization protocols

A 20% mucin solution (cat# M1778, Sigma Aldrich, Missouri, US) was prepared in a sterile 15 mL conical tube. After thorough mixing, the tube was placed in water bath for 2 hours at 55˚C. Consistency of mucin solution was checked at intervals of 30 min with naked eye and gentle shaking to ensure proper solubilization. The tubes were placed at room temperature (21–23˚C) for 4 hours for cooling, when mucin solution consistency was checked for its resemblance to sputum consistency. The solution was divided into 500 μL aliquots and stored at 2–8˚C in refrigerator for future use. The initial viscosity of porcine mucin was rated as ++++, and then from ++++ (most viscous, or unaltered) to + (least viscous, or liquefied) after the chemical treatments. Viscosity was observed and subjectively measured with the naked eye while gently agitating the tubes. Relative viscosity was performed by two researchers (NA and ADTC) in three independent experiments per researcher. Eight different compositions of the liquefying solution, also called "Solution 01" hereafter, were evaluated (Table 1). The details for preparation of each solution are outlined below and on Table 1. NALC/NaOH solution was prepared by mixing equal volumes of 6% NaOH (3 grams NaOH in 50 mL distilled water) and 2.9% sodium citrate (1.45 grams Na-citrate in 50 mL distilled water). The solution was sterilized by autoclaving at 121˚C for 15 minutes and kept in the refrigerator until needed, according to WHO protocols [6]. Immediately before use, 0.5% (i.e., 0.25 grams) N-acetyl-L-cysteine (cat# A7250, Sigma Aldrich, Missouri, USA) was added to 50 mL of NALC/NaOH and dissolved by gentle inversion. The resulting solution was stored at 2–8˚C until 15 minutes prior to use, when it should be warmed up to room temperature [6]. GSCN solution (6 M) was prepared by dissolving 17.72 grams of guanidine thiocyanate (GSCN, cat# G9277, Sigma Aldrich, Missouri, USA) in 10 mL of a 0.5 M EDTA solution (pH 8.0) in a 50 mL conical tube. The tube was heated at 65˚C for 1 hour with occasional inversion to help homogenization. After incubation, volume was brought up to 25 mL with 0.5 M EDTA (pH 8.0) and the total volume was divided into 500 μL aliquots. Urea solution (8 M) was prepared by dissolving 48.05 grams of urea (cat# U5378, Sigma Aldrich, Missouri, USA) in 60 mL of distilled water. After the reagent was completely dissolved, the volume was adjusted to 100 mL with distilled water. Triton X-100 (10%) was prepared by diluting 5 mL of 100% Triton X-100 (cat# T8787, Sigma Aldrich, Missouri, USA) in 45 mL distilled water. SDS (10%) was prepared by dissolving 0.5 grams of sodium dodecyl sulphate (cat# L3771, Sigma Aldrich, Missouri, USA)

**Table 1. Liquefaction of mucin with different chemicals.**

| Tube # | Mucin | NALC/ NaOH | PBS | GSCN | Urea | TX100 / SDS / NP40 | Total | Liquefaction Before/After |
|---|---|---|---|---|---|---|---|---|
| [Stock] | 20% (w/v) | 0.5% / 3% (w/v) | 100 mM | 6 M | 8 M | 10% | - | - |
| 1 | 500μL | 500μL | 500μL | - | - | - | 1500μL | ++++ / + |
| 2 | 500μL | 500μL | 500μL | 400μL | 400μL | 100μL | 2000μL | ++++ / + |
| 3 | 500μL | - | - | 200μL | 200μL | 100μL | 1000μL | ++++ / ++ |
| 4 | 500μL | - | - | 500μL | 500μL | 200μL | 1700μL | ++++ / ++ |
| 5 | 500μL | - | - | 250μL | 250μL | 100μL | 1100μL | ++++ / ++ |
| 6 | 500μL | - | - | 500μL | - | - | 1000μL | ++++ / + |
| 7 | 500μL | - | - | - | 500μL | - | 1000μL | ++++ / ++ |
| 8 | 500μL | - | - | - | - | 200μL | 700μL | ++++ / +++ |
| 9 | 500μL | - | - | 300μL | 100μL | 50μL | 1000μL | ++++ / ++ |

Chemicals and their respective concentration and volumes used to produce different liquefying solutions. Porcine mucin was used as sputum substitute and its viscosity was classified with the naked eye by gentle agitation of the tube. Viscosity of each solution was classified from the most viscous (++++) to less viscous (+).

in 50 mL distilled water. Nonidet P-40 substitute, or NP-40, (10%) was prepared by diluting 5 mL of 100% NP-40 substitute (cat# 74385, Sigma Aldrich, Missouri, USA) in 45 mL distilled water. All surfactant solutions were homogenized by gentle inversion to avoid formation of bubbles. Hydrochoric acid (HCl, cat# H1758, Sigma Aldrich, Missouri, USA) was diluted from the concentrated stock (36.5–38%, approximately 36 M) to 1 M in distilled water and stored at room temperature. Solution 02 consisted of commercial solution of Tris-EDTA (TE) pH 8.0 (cat# AM9856, Ambion, Massachusetts, USA). Except where noted, all solutions were stored at room temperature.

## Sample preparation and application onto FTA card

One mL of the sputum sample was mixed and thoroughly homogenized with 400 μL of the liquefying solutions (Table 1) for 30 seconds in vortex. The "Solution 01" with the best liquefying ability (GSCN 6M), as shown in Table 1 (tube #6), was used in all the next steps. This solution, GSCN 6M, will be referred to as Solution 01 for the remaining steps of the study. The homogenized mixture ("sample plus Solution 01") was uniformly applied onto a FTA ELUTE Micro Card (cat# WB120401, GE Health Care Life Sciences, Chicago, US). The card was allowed to dry for 1 hour at room temperature (21–23˚C).

## DNA elution protocols

Six DNA elution protocols were evaluated (Table 2). The FTA ELUTE Micro Card has four circular pre-marked areas, each with 11 mm of diameter. Either one entire circular area or one disc of 6 mm in diameter were punched off the FTA card that was spotted with sample. The punched discs were collected in a clean 1.5 mL tube. Five hundred microliters (500 μL) of Solution 02 were added to each tube, which was either vortexed for 3 times of 5 seconds with 5-seconds intervals or sonicated in water bath for 3 times of 15 seconds with 5-seconds intervals. Next, tubes were incubated at room temperature or 95˚C for 5 minutes. Tubes were cooled down to room temperature, and the supernatants containing the extracted DNA were collected into a new tube. The supernatant was used for qPCR analyses or stored at -20˚C for future use. The remaining of the FTA ELUTE Micro Card was stored at room temperature for future use. For protocol optimization, DNA from the each sample was extracted 2–3 times for each experimental condition. The concentration of the eluted DNA was measured spectrophotometrically at 260 nm on a Nanodrop 2000 (Thermo Fisher Scientific, USA).

**Table 2. Steps of each of the six different DNA elution protocols evaluated in the present work.**

| Protocol 1 | Protocol 2 | Protocol 3 | Protocol 4 | Protocol 5 | Protocol 6 |
|---|---|---|---|---|---|
| 1. Take the whole circle (11 mm) | 1. Take the whole circle (11 mm) | **1. Take one punch (6 mm)** | 1. Take one punch (6 mm) | 1. Take the whole circle (11 mm) | 1. Take one punch (6 mm) |
| 2. Add 500μL $H_2O$ or TE pH 8.0 | 2. Add 500μL $H_2O$ or TE pH 8.0 | **2. Add 500μL $H_2O$ or TE pH 8.0** | 2. Add 500μL $H_2O$ or TE pH 8.0 | 2. Add 500μL $H_2O$ or TE pH 8.0 | 2. Add 500μL $H_2O$ or TE pH 8.0 |
| 3. Vortex 3x for 5 sec | 3. Vortex 3x for 5 sec | **3. Vortex 3x for 5 sec** | 3. Vortex 3x for 5 sec | 3. Water bath sonication (3x 15 sec) | 3. Water bath sonication (3x 15 sec) |
| 4. Incubate at 95˚C /5 min | 4. Room temperature / 5 min | **4. Incubate at 95 ˚C / 5 min** | 4. Room temperature / 5 min | 4. Room temperature / 5 min | 4. Incubate at 95˚C / 5 min |
| 5. Collect supernatant | 5. Collect supernatant | **5. Collect supernatant** | 5. Collect superrnatant | 5. Collect supernatant | 5. Collect supernatant |
| 6. Run qPCR | 6. Run qPCR | **6. Run qPCR** | 6. Run qPCR | 6. Run qPCR | 6. Run qPCR |

Each protocol was devised to evaluate one or a combination of chemical and physical properties of each step to maximize elution of DNA from the FTA cards. In bold, the protocol that yielded the best results (protocol #3).

## Limit of Detection (LOD): Colony forming units in parallel to qPCR

Reference strain *M. tuberculosis* H37Rv colonies cultivated in Ogawa-Kudoh medium were collected and homogenized with glass beads. The turbidity of the bacterial solution was compared to the turbidity of the McFarland number 1 standard (widely used in microbiology, especially for conventional anti-biograms). Subsequently, ten-fold serial dilutions were performed from the concentrated cell suspension, and individual mucin samples of 500 μL were spiked with 30 μL of each dilution. The whole volume was equally distributed onto individual FTA ELUTE Micro Cards after incubation with Solution 01 (see "Sample preparation and application onto FTA card"). The procedure was performed in duplicate for each concentration. The most efficient DNA extraction protocol as determined in section "DNA elution protocols" (protocol #3, in bold in Table 2), in parallel with the second best (protocol #4), was performed on each FTA card, and qPCR assays on the ABI7500 or the Q3-Plus system were used to evaluate the Ct for each dilution. The same procedure was performed in parallel and spiked mucin samples were plated in duplicates in petri dishes containing 7H10/OADC medium. Colonies were counted using a semi-quantitative scale, according to the guidelines of the Brazilian Surveillance Laboratory for Tuberculosis and other Mycobacteria [16].

## Gelification of qPCR reagents

Gelification of the qPCR reagents was performed by adding a gelification solution in substitution for water, according to published protocols The final concentration of each component of the gelification mixture used in this study was trehalose 160 mg/mL, melezitose 80 mg/mL, glycogen 40 mg/mL and lysine 0.15 mg/mL. all qPCR reagents (enzymes, buffers, oligonucleotides, dNTPs, etc.) and 5 μL of in substitution for molecular-grade water in [12, 17–19]. The gelification solution contains three classes of components, each with specific functions within the mixture and the process: (i) trehalose and melezitose (400 mg/mL each), to protect the biomolecules from the desiccation process by stabilizing and reducing the activity of water in the solution; (ii) lysine (0.75 mg/mL), a free radical scavenger which inhibits oxidizing reactions between carbonyl or carboxyl groups and the biomolecule's amino or phosphate groups; and (iii) glycogen (200 mg/mL), an inert polymer which creates a broader protective matrix. These components define a sol-gel mixture that prevent the loss of the tertiary or quaternary structure during the desiccation process, thus helping maintaining the biomolecules' activity upon rehydration [12, 17]. The final concentration of each component of the gelification mixture used in this study was trehalose 160 mg/mL, melezitose 80 mg/mL, glycogen 40 mg/mL and lysine 0.15 mg/mL. The qPCR mix containing all qPCR reagents (enzymes, buffers, oligonucleotides, dNTPs, etc.) and 5 μL of the gelification solution in substitution for molecular-grade water was manually aliquoted into each reaction well. Next, the reaction vessels (either the ABI7500 plastic 96-well plates or the Q3-Plus silicon chips) were submitted to vacuum (30 ± 5 mbar) under controlled temperature (30 ± 1˚C) to gelify the reagents. For all experiments, gelified reactions were stored at 2–8˚C for up to 7 days.

## Experimental conditions for qPCR

For detection of MTB, primers and probe for the *IS*6110 insertion element of the *Mycobacterium* complex were purchased as a custom Taqman® gene expression assay (Table 3) (Thermo Scientific, California, USA). Regular or gelified reactions were performed on a standard ABI7500 instrument (Applied Biosystems) or on the portable prototype instrument Q3-Plus [20]. The Q3-Plus system has small dimensions (7 × 14 × 8.5 cm), weighs only 300 grams, performs the thermal cycles in a disposable silicon chip, and can be operated with any small portable computer. The reactions contained the Multiplex PCR Mastermix (IBMP,

**Table 3. Sequence of primers and probes used for detection of the genomic MTB target IS6110 and of the human gene 18S rRNA.**

| Name | Sequence (5´-3´) |
|---|---|
| IS6110 Fw | CAGGACCACGATCGCTGAT |
| IS6110 Rv | TGCTGGTGGTCCGAAGC |
| IS6110 Probe | FAM-TCCCGCCGATCTCG-MGB |
| 18S Fw | TGCGAATGGCTCATTAAATC |
| 18S Rv | CGTCGGCATGTATTAGCTCT |
| 18S Probe | VIC-TGGTTCCTTTGGTCGCTCGCT-BHQ1 |

Curitiba, Brazil) and MTB Taqman Oligomix (assay name IS6110-A, assay ID: AIPACNK, Thermo Scientific, USA). Regular and gelified reactions followed the same cycling conditions as well as analysis parameters. In the ABI7500 instrument, reactions were standardized to a final volume of 25 μL, considering the addition of 5 μL of sample (extracted DNA), and the cycling conditions were 50˚C for 2 minutes, 95˚C for 10 minutes, followed by 45 cycles of 95˚C for 15 seconds and 60˚C for 60 seconds. Baseline and threshold were set to automatic. In the Q3-Plus instrument, 1 μL of the extracted DNA was added to a final volume of 5 μL, and the cycling conditions were: 80˚C for 10 seconds, 97˚C for 60 seconds, followed by 45 cycles of 97˚C for 20 seconds and 62˚C for 60 seconds. The optical parameters for the FAM channel in the Q3-Plus system were exposure time of 1 second, led power of 3 and analog gain of 15, while for the VIC channel the optical parameters were exposure time of 2 seconds, led power of 8, and analog gain of 14. Baseline is automatic and threshold was set to 15 arbitrary units of fluorescence. Reactions on both instruments were also supplemented with oligonucleotides for the detection of the human 18S rRNA gene, as an internal reaction control (Table 3). The detection of a human gene confirms the presence of the sample in the reaction, as well as controls for the presence of possible contaminants or inhibitors [11, 21].

## Statistical analysis

Reactions on the ABI7500 instrument were performed in triplicates, and reactions on the Q3-Plus instrument were performed in duplicates (optimization protocols) or quadruplicates (patient samples). The Cohen's kappa coefficient was calculated between the Q3-Plus or ABI7500 results obtained with the chosen protocol and the results obtained with GeneXpert or culture, using these latter methods as the gold standard with a 95% CI (confidence interval). The coefficient was used to test the agreement between the diagnostic methods/instruments, and Kappa results were interpreted according to [22]: 1.00–0.81 almost perfect, 0.80–0.61 substantial, 0.60–0.41 moderate, 0.40–0.21 fair and ≤0.20 slight agreement. Student t-test with a significance level of 0.05 was used to evaluate the difference between results obtained with the ABI7500 and the Q3-Plus system. The LOD for the DNA extraction protocols was estimated by qPCR in parallel to colony growth.

## Results

### Sample liquefaction and protocol optimization

We hypothesized that concentrated GSCN and other chemicals would liquefy a sputum-like matrix consisting of porcine mucin. GSCN (6 M) was used alone, or mixed with other chaotropic or surfactant chemicals, such as urea (8 M), Triton X-100 (10%), SDS (10%), or NP-40 (10%). Several volumes and combinations of chemicals were evaluated (Table 1). Viscosity was observed and subjectively measured with the naked eye while gently agitating the tubes by two researchers in three independent experiments. Relative liquefaction results

shown in Table 1 represents the average frequency observed in the six experiments. The "Control" sample (tube #1) was treated according to the digestion-decontamination protocol suggested by WHO with small modifications such as not using phosphate buffer to stop the decontamination process because preservation of *M. tuberculosis* bacilli for culturing is not the purpose of the present protocol. Hence, NALC/NaOH was added to Tube #1, resulting in complete liquefaction (+). Tube #2 tests if any of the chemicals that will be evaluated would interfere with the standard procedure (NALC/NaOH), and no interference was observed. Triton X-100, SDS, or NP-40 alone or in combination were the least effective chemicals and showed little to no mucin-liquefying power (+++). Urea alone was more effective (++) than in combination with any surfactant (Triton X-100, SDS, or NP-40)(+++), but still without achieving the same level of liquefaction elicited by NALC/NaOH. The only chemical as effective as the control treatment was GSCN (+), either alone or in combination with urea or any of the surfactants (Table 1).

Next, we evaluated six different extraction protocols (Table 2). The protocols varied on punch size (one 6 mm punch, or the whole 11 mm circle), washing and elution buffers (H$_2$O or TE pH 8.0), as well as different incubation conditions (with or without incubation/sonication/vortexing). Mucin samples were spiked with 25 ng/µL of DNA extracted from H37Rv MTB cells. Protocols #3 and #4 showed better results when compared to the others (#1, #2, #5, or #6), as measured by amplification of the IS6110 genomic target on the standard benchtop equipment ABI7500 (Fig 2). Similar result was obtained with the portable system Q3-Plus. Interestingly, using the undiluted extracted DNA resulted in no amplification in neither instrument (Table 4). However, diluting both extracted DNA at 1:10 or 1:100 ratios resulted in positive amplifications, suggesting that some contaminants and inhibitors were present in the undiluted extract (Fig 1 and Table 4). Indeed, the difference of approximately 3–4 Ct values between the 1:10 and 1:100 diluted samples suggest that no inhibitor is present after the dilution. Interestingly, the same pattern was observed with the ready-to-use reactions performed in the portable qPCR instrument. Even though the detection occurred in a later Ct, the *IS*6110 marker was detected with the expected difference of approximately 3.3 Ct between the 1:10 and 1:100 dilutions. Table 4 summarizes the Ct values obtained for each dilution in protocols #3 and #4. A Student's t-

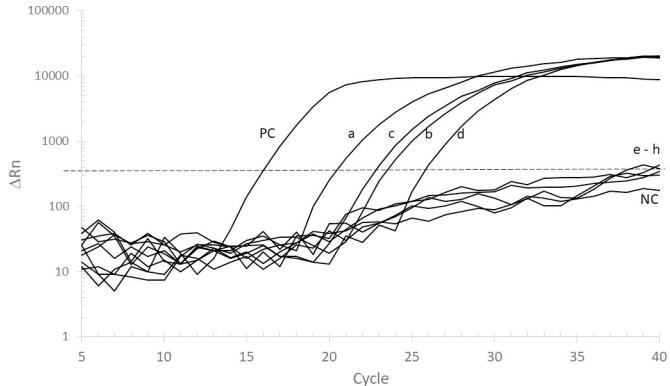

**Fig 2. Representative traces of the qPCR amplification of *M. tuberculosis IS*6110 target.** Porcine mucin samples spiked with *M. tuberculosis* H37Rv DNA (25 ng/µL) were processed according to protocols #3 or #4. Traces "a" and "b" were obtained after 1:10 and 1:100 dilution of the eluted DNA obtained with protocol #3. Traces "c" and "d" were obtained after 1:10 and 1:100 dilution of the eluted DNA obtained with protocol #4. "PC" and "NC" represent the positive control (25 ng/µL of DNA extracted from H37Rv MTB cells) and the negative control (TE pH 8.0), respectively. Traces "e", "f", "g", and "h", ("e–h") represent 1:100 dilutions of protocols #1, #2, #5, and #6. Traces are representative of at least three independent extraction procedures.

**Table 4. Comparison of ABI7500 and Q3 using protocols #3 and #4.**

| Protocol | Dilution | ABI7500 –Regular | Q3-Plus–Regular | Q3-Plus–Gelified |
|---|---|---|---|---|
| # 3 | No dilution | Undetermined | Undetermined | Undetermined |
| | 1:10 | 22.60 ± 0.36 | 22.10 ± 0.51 | 24.50 ± 0.91 |
| | 1:100 | 25.62 ± 0.60 | 25.43 ± 0.59 | 28.52 ± 0.74 |
| # 4 | No dilution | Undetermined | Undetermined | Undetermined |
| | 1:10 | 24.45 ± 0.54 | 24.57 ± 0.63 | 28.77 ± 1.09 |
| | 1:100 | 28.18 ± 0.52 | 28.53 ± 0.89 | 34.33 ± 1.16 |

The two best protocols were compared using a standard benchtop or a portable instrument. Undiluted supernatants did not yield any amplification, while simple 10-fold dilutions in TE pH 8.0 allowed the detection of *M. tuberculosis* DNA. Cts are shown as mean ± S.D.

test analysis of the observed means show that the results obtained with the Q3-Plus system are not significantly different from those obtained with the ABI7500 instrument using either protocol (p-value of 0.16 for protocol #3 and 0.42 for protocol #4, at the 1:10 dilution). Similarly, no significant difference was observed between the regular and the ready-to-use reagents (p-value of 0.22 for protocol #3 and 0.39 for protocol #4, at the 1:10 dilution). However, despite having a similar performance, the gelified results were statistically different than the ones obtained with the regular qPCR format (p-value of 0.02 for protocol #3 and 0.05 for protocol #4, at the 1:10 dilution) (Table 4). Taken together, these results suggest that protocol #3 yields the best results, and therefore it was used in the next experiments.

## Evaluation of FTA extraction protocol with intact MTB cells and colony forming unit counting

Starting from a *M. tuberculosis* suspension corresponding to the turbidity 1 on the McFarland scale, colony growth could be observed up to the fourth 1:10 dilution. Counting could be performed for the third and fourth dilutions, yielding an average of 15 ± 2.1 and 02 ± 0.7 CFU/ml, respectively (Table 5). PCR performed with the DNA extracted from the corresponding FTA cards using protocol #3 resulted in detection of the IS6110 target for all four dilutions, with the last detected dilution (fourth dilution) displaying a Ct of 33.95 ± 1.02 for the ABI7500 and of 35.90 ± 1.21 for the gelified reaction on the Q3-Plus (Table 5). Representative traces of the qPCR runs on the standard ABI7500 and the portable Q3-Plus instrument detecting DNA obtained from dilution $10^{-4}$ are shown in S1 Fig. The S1 Fig also shows a positive control (PC) for detection of 25 ng/μL of DNA extracted from H37Rv MTB cells. The efficiency of the qPCR detection was determined to be 105.4% (slope -3.20 and $R^2$ of 0.997) for the ABI7500 and 125.6% (slope -2.83 and $R^2$ of 0.987) for the Q3-Plus.

**Table 5. Colony forming unit and *IS*6110 detection by two qPCR instruments using 1:10 dilutions of a *M. tuberculosis* cell suspension.**

| Dilution | Average (CFU/mL) | ABI7500 (Ct ± SD) | Q3-Plus (Ct ± SD) |
|---|---|---|---|
| $10^{-1}$ | countless | 24.13 ± 0.78 | 27.45 ± 0.56 |
| $10^{-2}$ | countless | 27.37 ± 0.83 | 30.53 ± 0.70 |
| $10^{-3}$ | 15.5 | 30.02 ± 0.91 | 33.97 ± 0.83 |
| $10^{-4}$ | 1.5 | 33.43 ± 0.71 | 35.62 ± 0.82 |
| $10^{-5}$ | no growth | Undetectable | Undetectable |

Plates were seeded with the same volume and concentration of *M. tuberculosis* as FTA Micro Elute cards that were processed according to protocol #3.

## Evaluation of the proposed protocol using MTB patients' samples

All previous results shown here were produced using a concentrated porcine mucin suspension, which has sputum-like consistency [15]. Since sputum contains other substances that might interfere with the qPCR, such as proteases, nucleases, and blood, we sought to evaluate the proposed protocol using patients' samples. A small convenience sample of 17 patients previously characterized by GeneXpert PCR and culture was selected from a larger cohort.

The untreated samples were submitted to protocol #3, and the extracted DNA was evaluated on the standard (ABI7500) and portable (Q3-Plus) instruments (Table 6). Representative qPCR traces for the analysis of the extracted DNA on the ABI7500 and Q3-Plus instruments are shown in S2 Fig. Panel A shows the detection of the *Mycobacterium* IS6110 target, the human 18S rRNA target, and a positive control DNA from H37Rv MTB cells ("PC") on the standard instrument ABI7500. Panel B shows a similar qPCR run on the portable Q3-Plus instrument. A simple characterization as "Positive" or "Negative" for the presence of MTB DNA shows that the new protocol produces the same results as the GeneXpert, with only one sample producing a different result (sample #10). As expected, when GeneXpert results differ from culture, results obtained with protocol #3 differed too (samples #1 and #11). The calculated kappa coefficients show a substantial agreement (88.9%, Cohen's kappa of 0.77) between the Q3-Plus (or the ABI7500) and the GeneXpert sample characterization, and a moderate agreement (78.9%, Cohen's kappa of 0.58) between the Q3-Plus (or the ABI7500) and the culture results. Overall, when compared to GeneXpert, the Q3-Plus (or the ABI7500) showed a sensitivity of 90.0% ($CI_{95\%}$ 55.5–99.8%), and a specificity of 100% ($CI_{95\%}$ 59.1–100.0%). When compared to culture, the Q3-Plus (or the ABI7500) showed a sensitivity of 100% ($CI_{95\%}$ 59.0–100.0%), a specificity of 80.0% ($CI_{95\%}$ 44.4–97.5%). Most importantly, all detections of the human 18S rRNA were within acceptable ranges ($13 < Ct > 33$, [21]). The concentration of

**Table 6. Comparison of detection of *M. tuberculosis* among different platforms.**

| Sample # | Culture | GeneXpert | ABI7500 | Q3-Plus |
|---|---|---|---|---|
| 1 | Negative | Positive | Positive | Positive |
| 2 | Positive | Positive | Positive | Positive |
| 3 | Negative | Negative | Negative | Negative |
| 4 | Positive | Positive | Positive | Positive |
| 5 | Negative | Negative | Negative | Negative |
| 6 | Negative | Negative | Negative | Negative |
| 7 | Positive | Positive | Positive | Positive |
| 8 | Positive | Positive | Positive | Positive |
| 9 | Positive | Positive | Positive | Positive |
| 10 | Negative | Positive | Negative | Negative |
| 11 | Negative | Positive | Positive | Positive |
| 12 | Positive | Positive | Positive | Positive |
| 13 | Negative | Negative | Negative | Negative |
| 14 | Positive | Positive | Positive | Positive |
| 15 | Negative | Negative | Negative | Negative |
| 16 | Negative | Negative | Negative | Negative |
| 17 | Negative | Negative | Negative | Negative |

Selected human samples previously characterized by culture or the benchtop instrument GeneXpert were selected for evaluation using the proposed protocol #3. Reactions on the ABI7500 were performed in triplicates, and on the Q3-Plus in quadruplicates. Characterization as "positive" means at least two of the replicate reactions were positive.

DNA in the eluates produced by applying protocol #3 to the patients' samples was $36.22 \pm 17.17$ ng/µL ($CI_{95\%}$ 30.53–41.91 ng/µL).

## Discussion

In this study, our aim was to demonstrate a proof of concept protocol for a rapid, simple, and cost-effective sample preparation for DNA extraction from samples containing *M. tuberculosis.* The protocol comprises a paper-based DNA extraction and specific target amplification/detection by a portable and easy-to-use real time PCR platform. An overview of the full procedure is outlined in Fig 3, showing a direct comparison to the general workflow of commercially available spin-column extraction kits. Although the ASSURED requirements have been met to a large extent in platforms such as GeneXpert Ultra or BioFire, the Q3-Plus system shows some advantages over both the aforementioned instruments and the standard benchtop ABI7500 in terms of equipment and reaction costs, reaction volume, user friendly interface, fast temperature ramping, and shorter total reaction time [12, 20]. Some of these features will be further discussed below.

For efficient diagnosis of TB, one must liquefy the sputum to disrupt the complex network of interlinked mucin proteic matrix, which is the main responsible for trapping the target microorganism [23]. Different strategies (chemical, mechanical or a combination of both) can be applied for sputum liquefaction, in a process where it changes from viscous and heterogeneous to a homogenous and liquefied state [4]. In the current study, several chemicals were screened for liquefaction and homogenization of sputum samples such as the chaotropic agents GSCN and urea, or the detergents NP-40, SDS, and Triton X-100. In previous studies, Triton X-100, urea, and GSCN were shown to eliminate all bacteria in sputum samples, including *M. tuberculosis* [8, 24]. Interestingly, GSCN by itself showed very efficient results in terms of sputum liquefaction, even when combined with the other chemicals. It has been previously shown that concentrated GSCN can be used in sample preparation for the extraction of DNA and protein from a variety of samples [25], and even using an electricity-free protocol [26] or small battery-operated instruments [27]. More importantly, *M. tuberculosis* was shown to become non-viable after exposure to commercial lysis buffers, which contain concentrated guanidine [8]. This guanidine feature is essential to the protocol developed in the present

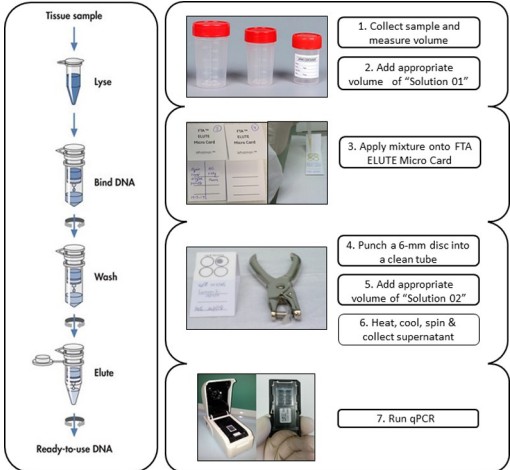

**Fig 3. Comparative flowchart of the proposed simplified protocol.** On the left panel, a general procedure for spin-column DNA extraction used in most commercial products. On the right panel, a step-by-step description of the simple and straightforward protocol for POC screening of tuberculosis-suspected samples.

work, since we propose the sample to be manipulated in low-resource settings, which might not have safety cabinets available for diagnostic routines. Moreover, we suggest that the current protocol should use the whole sample volume in the initial liquefaction step, in order to avoid the known heterogeneity of the sputum samples. In such scenario, samples would be immediately decontaminated by the concentrated guanidine isothiocyanate and *M. tuberculosis* would be made inviable, presenting no risks for the operator in the subsequent steps. The FTA ELUTE Micro Card, based on Whatman FTA technology, are chemically treated paper cards that enable cell lysis and release of nucleic acids upon contact [28]. Integrating liquefaction and decontamination of the sputum sample to a paper-based DNA extraction is another approach for TB diagnosis adopted in our proposed protocol. Others have demonstrated a rapid and cost-effective sample preparation using a microfluidic "origami", using spiked mucin [15] or even artificial sputum [27] as models. When compared to the microfluidic "origami", the FTA card is more efficient in cell lysis and disintegration of the viscous matrix of a sputum sample because of the dry reagents are already stored on the paper. Moreover, the microfluidic "origami" procedure takes around 2 hours while the protocol described here takes 1.5 hours from raw sample to eluted DNA. The semi-automated PureLyse® protocol, although as fast as our protocol, does not show the same sensitivity [27]. Interestingly, the use of FTA cards for storage, transport and preservation of sputum samples has been used successfully for diagnosis of MTB [29, 30]. These authors have shown that MTB DNA was stable in the FTA card for six months at room temperature [29], and that the extracted DNA is still suitable for PCR [30]. The findings provide not only a simple and economically favorable solution for collection, storage and transportation of the sputum but also suggest the use of these cards for sample preparation and DNA extraction in low-income countries such as the protocol described in the present work.

In the present study, we chose to use IS6110, a *M. tuberculosis* complex genomic marker. This target is widely used in diagnostic and epidemiologic studies due to high number of copies (up to 25) and high discriminatory ability. Although some studies have shown M. tuberculosis isolates with lower number of IS6110 copies, this genomic marker is considered one of the best options to increase the sensibility of diagnostic tests. Indeed, the inclusion of the genomic sequences IS6110 and S1081 in the new generation of the GenXpert MTB/Rif test resulted in increased sensibility [31–36].

The low number of samples analyzed in the present study limits our ability to draw definitive conclusions, but the results obtained with the optimized protocol showed good agreement with results obtained by reference diagnosis methods such as culture or other molecular tests, providing the basis for future studies. The ready-to-use reactions performed similarly to the conventional ones, exhibiting similar lower detection limits, as previously shown for detection of *T. cruzi* or *Plasmodium* spp DNA [12]. After evaluation of larger cohorts, we suggest that the ready-to-use reactions could be implemented in places with unreliable power supply, as in urban suburbs and rural areas of high TB burden countries of Brazil, India, Indonesia, China, Philippines and Pakistan. The insertion of an internal control in the reaction, i.e. the detection of the human 18S rRNA gene, assures the functionality of the system regarding sample extraction, nucleic acid integrity, the absence of inhibitors in the reaction, and correct functioning of equipment and software [12, 21].

For comparison purposes, the cost of DNA extraction with a commercial kit is about US $3–5 per sample, while the cost of our extraction protocol using GSCN+FTA paper can be roughly estimated at US$1–2 per sample. The cost of the PCR reaction per patient using the Q3-plus instrument and ABI 7500 is similar (US$3–4) while there is a significant difference in the cost of the instrument (Q3-plus <US$10,000 while ABI7500 >US$40,000) [12, 20]. In general, the ABI7500 or any other benchtop instrument needs skilled professionals, a sophisticated

lab environment and routine maintenance and calibration, while the Q3-Plus platform relies on minimum personal training and does not require maintenance/calibration or any other lab infrastructure.

Turnaround time is also an important aspect of any diagnostic assay. The culture technique, which is the standard for MTB diagnosis, can take from 4 to 16 weeks [37], and such a long period can result in further complexities such as delays in initiation of the treatment or loss of disease follow up. On the other hand, GeneXpert Ultra have turnaround time of 80–90 min but have relatively high cost per sample (concessional price with up to 90% discount is US $9.98 for eligible countries). We have estimated the turnaround time for our tests as 3 hours from raw sample to result. This feature provides a rapid diagnosis that, thanks to the accurate molecular detection elicited by qPCR, leads to an early and more efficient start of treatment. Our assay reagents are workable at room temperature without the need for any special storage infrastructure, which makes tuberculosis diagnosis feasible in poorly equipped areas. Furthermore, access to quality diagnosis at POC will not only allow timely initiation of treatment but also reduce the dropout rate of the patients, which is one of the key factors for the spread of TB [2].

It should be stressed that the results are presented herein as a proof of concept, and that the protocol and reported LOD must be evaluated in larger cohorts. Another important aspect of the proof of concept protocol is that it was developed using an artificial matrix, as other have already done [15]. However, when evaluated with real sputum samples, the results were not different from those obtained with a reference molecular method (Table 6), suggesting that any possibly present inhibitors did not impair the qPCR detection of *M. tuberculosis* DNA.

## Conclusions

The present work shows the development of a simple and inexpensive sample preparation protocol for the detection of TB in viscous sputum sample, integrated with an easy-to-use portable real time PCR system, thus comprising a simple sample-to-answer diagnostic tool. Our sample preparation protocol allows the detection of a MTB bacterial load as low as 02 CFU/mL. The work presented here is a step towards a point of care solution for MTB diagnosis, suitable for low infrastructure primary health units as well as secondary healthcare clinics.

## Supporting information

**S1 Fig. Representative traces of the qPCR detection of DNA extracted from the smallest dilution that produced countable colonies.** Panel A, trace obtained on the standard ABI7500 with DNA from dilution $10^{-4}$. Panel B, trace obtained on the portable Q3-Plus with DNA from dilution $10^{-4}$.
(PDF)

**S2 Fig. Representative traces of the qPCR detection of DNA extracted from a positive patient sample.** Shown is the detection of the positive control (PC) or the IS6110 target (blue lines) and the human 18S target (red lines). Panel A, trace obtained on the standard ABI7500. Panel B, trace obtained on the portable Q3-Plus.
(PDF)

**S1 Table. Data used to calculate the averages presented on Table 4.**
(DOCX)

**S2 Table. Data used to calculate the averages presented on Table 5.**
(DOCX)

## Acknowledgments

The authors would like to thank Dr. Thiago Jacomasso for valuable input during writing of the manuscript, and Mr. Nilson Fidêncio and Ms. Silvia Bohn for the excellent technical assistance.

## Author Contributions

**Conceptualization:** Maria Lúcia Rosa Rossetti, Alexandre Dias Tavares Costa.

**Data curation:** Nasir Ali, Graziele Lima Bello, Maria Lúcia Rosa Rossetti, Alexandre Dias Tavares Costa.

**Formal analysis:** Nasir Ali, Graziele Lima Bello, Maria Lúcia Rosa Rossetti, Alexandre Dias Tavares Costa.

**Funding acquisition:** Maria Lúcia Rosa Rossetti, Marco Aurelio Krieger, Alexandre Dias Tavares Costa.

**Investigation:** Nasir Ali, Graziele Lima Bello.

**Methodology:** Nasir Ali, Graziele Lima Bello, Maria Lúcia Rosa Rossetti, Alexandre Dias Tavares Costa.

**Project administration:** Maria Lúcia Rosa Rossetti, Marco Aurelio Krieger, Alexandre Dias Tavares Costa.

**Resources:** Marco Aurelio Krieger, Alexandre Dias Tavares Costa.

**Supervision:** Maria Lúcia Rosa Rossetti, Marco Aurelio Krieger, Alexandre Dias Tavares Costa.

**Validation:** Graziele Lima Bello, Maria Lúcia Rosa Rossetti, Alexandre Dias Tavares Costa.

**Writing – original draft:** Nasir Ali.

**Writing – review & editing:** Graziele Lima Bello, Maria Lúcia Rosa Rossetti, Marco Aurelio Krieger, Alexandre Dias Tavares Costa.

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
