## [Decision Letter · Decision Letter 0]

28 Aug 2020

PONE-D-20-24367

Demonstration of a fast and easy sample-to-answer protocol for tuberculosis screening in point-of-care settings: a proof of concept study §

PLOS ONE

Dear Dr. Costa,

Thank you for submitting your manuscript to PLOS ONE. After careful consideration, we feel that it has merit but does not fully meet PLOS ONE’s publication criteria as it currently stands. Therefore, we invite you to submit a revised version of the manuscript that addresses the points raised during the review process.

We look forward to receiving your revised manuscript.

Kind regards,

Hasnain Seyed Ehtesham

Academic Editor

PLOS ONE

Journal Requirements:

2. We note that you state "Various ratios of sample to the liquefying solutions were evaluated, and the best results were obtained by mixing one volume of sputum sample to 0.4 volumes of Solution 01 (not shown)." Unfortunately, reference to unpublished or unshown data does not meet our data availability criteria (https://journals.plos.org/plosone/s/data-availability). Please either provide this data in the manuscript or a public repository, or if the data is not essential for your study, you may remove it.

"This work was partially funded by a grant from the Banco Nacional de

Desenvolvimento Econômico e Social (BNDES - bndes.gov.br), contract no.

15.2.0473.1 (Operation #4.816.864) to ADTC. The funders had no role in study design,

data collection and analysis, decision to publish, or preparation of the manuscript."

"I have read the journal's policy and the authors of this manuscript have the following

competing interests: authors NA, MAK, and ADTC are affiliated with Instituto de

Biologia Molecular do Paraná (IBMP), producer of the PCR mastermix used in the

qPCR experiments."

6. Your ethics statement must appear in the Methods section of your manuscript. If your ethics statement is written in any section besides the Methods, please move it to the Methods section and delete it from any other section. Please also ensure that your ethics statement is included in your manuscript, as the ethics section of your online submission will not be published alongside your manuscript.

7. We note that your paper includes detailed descriptions of individual patients (In figure 2). As per the PLOS ONE policy (http://journals.plos.org/plosone/s/submission-guidelines#loc-human-subjects-research) on papers that include identifying, or potentially identifying, information, the individual(s) or parent(s)/guardian(s) must be informed of the terms of the PLOS open-access (CC-BY) license and provide specific permission for publication of these details under the terms of this license. Please download the Consent Form for Publication in a PLOS Journal (http://journals.plos.org/plosone/s/file?id=8ce6/plos-consent-form-english.pdf). The signed consent form should not be submitted with the manuscript, but should be securely filed in the individual's case notes. Please amend the methods section and ethics statement of the manuscript to explicitly state that the patient/participant has provided consent for publication: “The individual in this manuscript has given written informed consent (as outlined in PLOS consent form) to publish these case details”.

Additional Editor Comments (if provided):

Major Revision

Reviewers' comments:

Reviewer's Responses to Questions

**Comments to the Author**

1. Is the manuscript technically sound, and do the data support the conclusions?

Reviewer #1: Yes

Reviewer #2: No

2. Has the statistical analysis been performed appropriately and rigorously? 

Reviewer #1: N/A

Reviewer #2: Yes

3. Have the authors made all data underlying the findings in their manuscript fully available?

Reviewer #1: Yes

Reviewer #2: No

4. Is the manuscript presented in an intelligible fashion and written in standard English?

Reviewer #1: Yes

Reviewer #2: Yes

5. Review Comments to the Author

Reviewer #1: • Authors need to thoroughly rewrite methodology of the study for more clarity.

• Authors should clearly mention the patient’s enrollment in the study whether patients were new cases or previously treated cases, exclusion and inclusion criteria of the patients enrolled, in methodology section.

• Line no. 176, authors need to clearly define the term “non-TB patients”.

• Authors should clearly mention the volume of sputum sample collected from each patient and clearly state whether that the informed consent has been taken from each patient.

• According to the WHO guidelines for TB diagnosis, sputum samples are needed to be collected for two consecutive days. In the present study, how the sputum samples have been collected? Whether these samples were from day 1 or day 2 or pooled? The author should clearly mention the collection of sputum samples from the patients enrolled.

• Author should provide schematic diagram of study design for more clarity.

• The study has smaller number of samples to be compared to give any conclusion.

• Author may provide supplementary file of data for more clarity.

• Authors have chosen the Insertion Sequence IS6110 from Mycobacterium tuberculosis complex as genomic marker. However, insertion sequence IS6110 of MTB has limitation of variable number of copies from multiple to single in different isolates. Author should discuss this issue in the study.

• Authors should share the load of human DNA in each sample and the data obtained after RT-PCR and may provide supplementary file of data.

Reviewer #2: The manuscript “Demonstration of a fast and easy sample-to-answer protocol for tuberculosis screening in point-of-care settings: a proof of concept study” provides minor but important improvisation in the TB diagnostics. Their instrument free DNA sample preparation from sputum with long term stability may have wider application in the future. I have following suggestions

1. As presented in the table-1, qualitative determination of sample viscosity by visualisation should be performed by more than one researcher and classification with maximum frequency should be presented for better reliability.

2. PCR results from new paper-based DNA isolation vs classical DNA isolation should be performed using same set of patient samples and gel image should be given in the manuscript.

3. At line 185-189, specific and exact composition and protocol of the sample liquefication is not mention. As this is the whole basis of this manuscript, the said paragraph must be re-written for clarity and reproducibility. Just mentioning “Solution 01 (not shown)” is not sufficient.

4. Similar to the last point, modify the line 220-222, the ambiguity caused my mysterious solution-1 without specific details.

5. As per line 223-224, ‘The most efficient DNA extraction protocol as determined in section “DNA elution protocols’; the most efficient protocol must be identified and stated in the paragraph.

6. Impact of gelification clubbed with new DNA extraction should be compared with classical protocols and gel image should be attached in the manuscript.

7. In line 242-243, specific and complete description of final working gelification solution and protocol must be given.

8. In the discussion, line 454-459, the detail itself is incorrect and must be removed. As given in the manuscript, “This raises another question: since qPCR detects DNA sequences, dead bacilli would also be detected and, therefore, this screening tool is not suitable for detecting active TB infections. However, the presence of MTB DNA in the sample is indicative of close contact to a TB patient, if not a latent infection itself...... to the culture”. The point discussed is absurd.

9. None of the figures (gel photo of PCR band or graph of RT-PCT) from the experiments displays the sensitivity of new method and its comparison with old methods.

10. As asserted previously, this manuscript presents a new and minor improvisation in the method and TB detection. If exact compositions and protocols of different steps are not given then, the whole purpose of presenting the manuscript to scientific world will become futile, worthless and unpublishable.

11. Make improvisation in language for error-free writing with better consistency and flow.

6. PLOS authors have the option to publish the peer review history of their article (what does this mean?). If published, this will include your full peer review and any attached files.

Reviewer #1: No

Reviewer #2: No

---

## [Author Response · Author response to Decision Letter 0]

19 Oct 2020

Dear Editor and Reviewers,

 Please find below a detailed response to all points raised in the review process. We greatly appreciate the comments and suggestions and truly believe that the manuscript is improved. 

 Sincerely,

 Alexandre Costa

Journal Requirements:

Reply: We have made all efforts to ensure that the manuscript meets all PLOS ONE's style requirements. 

2. We note that you state "Various ratios of sample to the liquefying solutions were evaluated, and the best results were obtained by mixing one volume of sputum sample to 0.4 volumes of Solution 01 (not shown)." Unfortunately, reference to unpublished or unshown data does not meet our data availability criteria (https://journals.plos.org/plosone/s/data-availability). Please either provide this data in the manuscript or a public repository, or if the data is not essential for your study, you may remove it.

Reply: The data is not essential to the study and the sentence was removed. 

Reply: We have changed the sentence and all data are now contained in the manuscript. However, detailed GeneXpert MTB/Rif data beyond the classification “positive” or “negative” cannot be published due to restrictions imposed by the ethics committee since these data were obtained from patient charts. These data can be obtained upon request to the Ethics Committee of the Universidade Luterana do Brasil at the email comitedeetica@ulbra.br (lines 577-580).

Reply: Some data cannot be publicized and can be accessed by contacting the Ethics Committee of the Universidade Luterana do Brasil at the email comitedeetica@ulbra.br, as stated on reply to item #3. This information is now present in the text (lines 577-580). All data supporting the manuscript is contained in the manuscript, figures, tables and supplementary tables. 

"This work was partially funded by a grant from the Banco Nacional de Desenvolvimento Econômico e Social (BNDES - bndes.gov.br), contract no. 15.2.0473.1 (Operation #4.816.864) to ADTC. The funders had no role in study design, data collection and analysis, decision to publish, or preparation of the manuscript."

Reply: Funding statement has been amended in the cover letter, as requested. 

5. Thank you for stating the following in the Competing Interests section: "I have read the journal's policy and the authors of this manuscript have the following competing interests: authors NA, MAK, and ADTC are affiliated with Instituto de Biologia Molecular do Paraná (IBMP), producer of the PCR mastermix used in the qPCR experiments."

Reply: The requested statement has been added to the cover letter. 

6. Your ethics statement must appear in the Methods section of your manuscript. If your ethics statement is written in any section besides the Methods, please move it to the Methods section and delete it from any other section. Please also ensure that your ethics statement is included in your manuscript, as the ethics section of your online submission will not be published alongside your manuscript.

Reply: The ethics statement has been moved to Methods (lines 134-138). 

7. We note that your paper includes detailed descriptions of individual patients (In figure 2). As per the PLOS ONE policy (http://journals.plos.org/plosone/s/submission-guidelines#loc-human-subjects-research) on papers that include identifying, or potentially identifying, information, the individual(s) or parent(s)/guardian(s) must be informed of the terms of the PLOS open-access (CC-BY) license and provide specific permission for publication of these details under the terms of this license. Please download the Consent Form for Publication in a PLOS Journal (http://journals.plos.org/plosone/s/file?id=8ce6/plos-consent-form-english.pdf). The signed consent form should not be submitted with the manuscript, but should be securely filed in the individual's case notes. Please amend the methods section and ethics statement of the manuscript to explicitly state that the patient/participant has provided consent for publication: “The individual in this manuscript has given written informed consent (as outlined in PLOS consent form) to publish these case details”.

Reply: Figure 2 does not contain detailed descriptions of individual patients. The inscriptions within the figure do not pertain to any patient and are merely illustrative. 

Additional Editor Comments (if provided):

Major Revision

Reviewers' comments:

Reviewer's Responses to Questions

Comments to the Author

1. Is the manuscript technically sound, and do the data support the conclusions?

Reviewer #1: Yes

Reviewer #2: No

2. Has the statistical analysis been performed appropriately and rigorously?

Reviewer #1: N/A

Reviewer #2: Yes

3. Have the authors made all data underlying the findings in their manuscript fully available?

Reviewer #1: Yes

Reviewer #2: No

4. Is the manuscript presented in an intelligible fashion and written in standard English?

Reviewer #1: Yes

Reviewer #2: Yes

5. Review Comments to the Author

Reviewer #1: 

• Authors need to thoroughly rewrite methodology of the study for more clarity.

Reply: Some sections in Methods have been rewritten and more information has been added. 

• Authors should clearly mention the patient’s enrollment in the study whether patients were new cases or previously treated cases, exclusion and inclusion criteria of the patients enrolled, in methodology section.

• Line no. 176, authors need to clearly define the term “non-TB patients”.

• Authors should clearly mention the volume of sputum sample collected from each patient and clearly state whether that the informed consent has been taken from each patient.

Reply: Patient’s exclusion and inclusion criteria are now clearly stated (lines 127-130). Patients were enrolled by attendance to the Department of Thisiology and Leprosy of the city of Canoas (RS, Brazil). Informed consent was not required by the Ethics Committee since the samples would be discarded (lines 136-137). Non-TB patients were defined by negative results in culture and GeneXpert analyses (lines 132-134). Volume of sputum collected from each patient is now clearly stated as an inclusion criteria (line 129). 

• According to the WHO guidelines for TB diagnosis, sputum samples are needed to be collected for two consecutive days. In the present study, how the sputum samples have been collected? Whether these samples were from day 1 or day 2 or pooled? The author should clearly mention the collection of sputum samples from the patients enrolled.

Reply: Samples were collected in two consecutive days, according to WHO guidelines. Characterization by the GeneXpert MTB/Rif assay and culture were performed on day 1 samples. The remaining of the day 1 sample was pooled with the day 2 sample, which was then evaluated by the new paper-based protocol. This information is now shown in lines 130-132.

• Author should provide schematic diagram of study design for more clarity.

Reply: A schematic diagram of the study workflow is now presented as Figure 1. 

• The study has smaller number of samples to be compared to give any conclusion.

Reply: We agree with the reviewer that the study was conducted with a small number of samples, and this has been mentioned in the discussion (lines 519-522). This is one of the main reasons we specified in title and discussion that the present study is a proof of concept and not a validation study (lines 554-555). Accordingly, the study does not intend to provide a definitive side-by-side comparison of the methods, but rather to demonstrate the feasibility of the method, as a proof of concept. As stated on lines 519-522 and 554-555, we believe that the demonstration of the feasibility of the method warrants the design of a study on a larger cohort that will allow a definitive conclusion. 

• Author may provide supplementary file of data for more clarity.

Reply: A figure showing the qPCR traces of the lowest detectable concentration (dilution 10-4, Table 5) as well as representative traces from a TB-positive sample is provided as supporting data (S1 Fig). We will gladly provide any other data upon a more specific request. 

• Authors have chosen the Insertion Sequence IS6110 from Mycobacterium tuberculosis complex as genomic marker. However, insertion sequence IS6110 of MTB has limitation of variable number of copies from multiple to single in different isolates. Author should discuss this issue in the study.

Reply: We agree with the reviewer and have now discussed this issue (lines 512-518). 

• Authors should share the load of human DNA in each sample and the data obtained after RT-PCR and may provide supplementary file of data.

Reply: We have added information about the average of the DNA concentration obtained from the patient samples (lines 442-443), which was found to be 36.22 ± 17.17 ng/µL. As stated in Methods (lines 289 and 292), 1 or 5 µL of the extracted DNA (for the portable or the standard instrument) were added to the qPCR reactions, thus representing roughly 7.3 ng/µL total DNA per reaction. It should be noted, however, that the exact amount of human DNA present in each sample is not possible to be calculated, since we also do not know how much M. tuberculosis DNA is present. Representative traces of the qPCR runs on the standard ABI7500 and the portable Q3-Plus instruments are shown in Supplementary Figures. 

Reviewer #2: 

The manuscript “Demonstration of a fast and easy sample-to-answer protocol for tuberculosis screening in point-of-care settings: a proof of concept study” provides minor but important improvisation in the TB diagnostics. Their instrument free DNA sample preparation from sputum with long term stability may have wider application in the future. I have following suggestions

1. As presented in the table-1, qualitative determination of sample viscosity by visualisation should be performed by more than one researcher and classification with maximum frequency should be presented for better reliability.

Reply: We regret that this information was not present in the initial text. Qualitative determination of sample viscosity by visualization was performed by two researchers (NA and ADTC) in three independent experiments per researcher. The relative viscosity was similar in all six experiments. This information is now given in lines 160-163 and 329-332.

2. PCR results from new paper-based DNA isolation vs classical DNA isolation should be performed using same set of patient samples and gel image should be given in the manuscript.

Reply: All patient samples were analyzed by both methods, the new paper-based method as well as by the GeneXpert MTB/Rif test. We understand that the GeneXpert test represents the classical DNA isolation methods. We regret that we cannot supply the qPCR plots generated by the GeneXpert analysis because they are part of patients’ charts and were restricted by the Ethics Committee. In addition, we would like to mention that we did not perform conventional PCR on any of the samples, and therefore we are not able to supply any gel images. However, we have included a representative qPCR for a patient sample, obtained with both instruments evaluated by the new protocol (ABI7500 and Q3-Plus) (Supplementary Figure 2).

3. At line 185-189, specific and exact composition and protocol of the sample liquefication is not mention. As this is the whole basis of this manuscript, the said paragraph must be re-written for clarity and reproducibility. Just mentioning “Solution 01 (not shown)” is not sufficient.

Reply: The specific and exact composition of all eight solutions evaluated as “Solution 01” are described on lines 151-191, and Table 1. The paragraph was re-written. 

4. Similar to the last point, modify the line 220-222, the ambiguity caused my mysterious solution-1 without specific details.

Reply: We are sorry for the ambiguity and believe that it is now resolved by stating that the Solution 01 with the best liquefying ability is GSCN 6M, as shown in lines 201-204 and by the results in Table 1. 

5. As per line 223-224, ‘The most efficient DNA extraction protocol as determined in section “DNA elution protocols’; the most efficient protocol must be identified and stated in the paragraph.

Reply: We regret that this information was present only in the description of Table 2. The most efficient DNA extraction protocol was #3, and is now clearly mentioned on line 243-244.

6. Impact of gelification clubbed with new DNA extraction should be compared with classical protocols and gel image should be attached in the manuscript.

Reply: The impact of the gelification on the new DNA extraction protocol was compared with the regular qPCR format in the portable and the standard instrument ABI7500 on Table 4. It was found that the gelification has an impact of 3-4 Ct on the detection by qPCR, but without impact on the ability of the reaction to detect the presence of the genomic target. This shift has also been in other reactions using the same portable instrument when compared to the standard instrument ABI7500 (Rampazzo et al 2019). This information is now included in the discussion (lines 522-524). 

7. In line 242-243, specific and complete description of final working gelification solution and protocol must be given.

Reply: Final concentration of each component of the gelification mixture is now given on lines 267-269. Volume of the gelification mixture used in the qPCR mix is stated on line 270-271. 

8. In the discussion, line 454-459, the detail itself is incorrect and must be removed. As given in the manuscript, “This raises another question: since qPCR detects DNA sequences, dead bacilli would also be detected and, therefore, this screening tool is not suitable for detecting active TB infections. However, the presence of MTB DNA in the sample is indicative of close contact to a TB patient, if not a latent infection itself...... to the culture”. The point discussed is absurd.

Reply: The text was removed as requested. 

9. None of the figures (gel photo of PCR band or graph of RT-PCT) from the experiments displays the sensitivity of new method and its comparison with old methods.

Reply: The sensitivity of the new method was compared to culture using two instruments (see Table 5). Representative qPCR traces of the lowest detectable dilution (10-4, Table 5) are shown as Supplementary Figure 1. 

10. As asserted previously, this manuscript presents a new and minor improvisation in the method and TB detection. If exact compositions and protocols of different steps are not given then, the whole purpose of presenting the manuscript to scientific world will become futile, worthless and unpublishable.

Reply: The steps for the preparation of all solutions as well as their exact composition (lines 151-191, and Table 1), and the steps of each protocol (lines 208-222, and Table 2) evaluated are present in the manuscript. We kindly ask the reviewer to point to any information that may have escaped our revision of the manuscript so that this oversight can be corrected. 

11. Make improvisation in language for error-free writing with better consistency and flow.

Reply: The final manuscript was thoroughly revised for English spelling, grammar, and flow. 

6. PLOS authors have the option to publish the peer review history of their article (what does this mean?). If published, this will include your full peer review and any attached files.

Do you want your identity to be public for this peer review? For information about this choice, including consent withdrawal, please see our Privacy Policy.

Reviewer #1: No

Reviewer #2: No

Reply: We greatly appreciate the comments and suggestions of the reviewers, and regret their decision to not disclose their identities as part of the peer review process.

---

## [Editor Report · Decision Letter 1]

3 Nov 2020

Demonstration of a fast and easy sample-to-answer protocol for tuberculosis screening in point-of-care settings: a proof of concept study §

PONE-D-20-24367R1

Dear Dr. Costa,

We’re pleased to inform you that your manuscript has been judged scientifically suitable for publication and will be formally accepted for publication once it meets all outstanding technical requirements.

Kind regards,

Seyed Ehtesham Hasnain

Academic Editor

PLOS ONE

Additional Editor Comments (optional):

I have gone through this revised manuscript and also the Authors response to reviewers comments. There were several issues raised by the Reviewers and Authors have comprehensively addressed all the issues. A schematic diagram of the study workflow is now presented as Figure 1 by the Authors. Revised manuscript shows a new and minor improvisation in the method and TB detection. Authors have now clearly stated the exclusion and inclusion criteria. As pointed out by the Reviewers, Authors have now described the specific and exact composition of all eight solutions evaluated as “Solution 01” and re-written that paragraph. The manuscript has now been revised extensively including making corrections and also other issues. I recommend this manuscript for publication.
---

## [Editor Report · Acceptance letter]

16 Nov 2020

PONE-D-20-24367R1 

Demonstration of a fast and easy sample-to-answer protocol for tuberculosis screening in point-of-care settings: a proof of concept study ^§^

Dear Dr. Costa:

I'm pleased to inform you that your manuscript has been deemed suitable for publication in PLOS ONE. Congratulations! Your manuscript is now with our production department. 

Kind regards, 

on behalf of

Prof Seyed Ehtesham Hasnain 

Academic Editor

PLOS ONE